# Breeding French bulldogs so that they breathe well—A long way to go

**Eva-Marie Ravn-Mølby**[1☯], **Line Sindahl**[1☯], **Søren Saxmose Nielsen**[1], **Camilla S. Bruun**[1], **Peter Sandøe**[1,2], **Merete Fredholm**[1]*

**1** Department of Veterinary and Animal Sciences, University of Copenhagen, Frederiksberg C, Denmark,
**2** Department of Food and Resource Economics, University of Copenhagen, Frederiksberg C, Denmark

☯ These authors contributed equally to this work.
* mf@sund.ku.dk

**Data Availability Statement:** All relevant data are within the paper and its Supporting Information files.

**Funding:** We wish to thank the Danish Kennel Club for financial support provided to MF for the project.

## Abstract

Brachycephalic syndrome (BS) is a pathophysiological disorder caused by excessive soft tissue within the upper airways of short-nosed dog breeds, causing obstruction of the nasal, pharyngeal and laryngeal lumen, resulting in severe respiratory distress. As the prevalence of BS appears to be high among some of the affected breeds, there is an urgent need for breeding efforts to improve the health status of those dogs. In the present study, we evaluated correlations between morphometric and other phenotypic characteristics and BS in a population of 69 French bulldogs from Denmark to identify parameters that could serve as a basis for breeding against BS. Furthermore, the genetic variation was monitored to determine whether it would be possible to breed based on these characteristics without simultaneously causing a critical reduction in genetic variation. Six phenotypic characteristics were correlated with the Brachycephalic Syndrome Functional (BSF) score. Among the morphometric risk factors, nostril stenosis (NS) and neck girth (NG) had the highest impact on the BSF score, accounting for 32% and 4% of the variation, respectively. The genetic variation in the population was comparable to other pure breeds, i.e. estimated and observed heterozygosity was 0.60 and the average inbreeding coefficient was 0.01. If only dogs with Grades 1 and 2 NS (no or only mild NS) were selected for breeding the mean BSF score would be reduced significantly. However, it would result in the exclusion of 81% of the population for breeding and this is not prudent. Excluding only dogs with severe stenosis (Grade 4) would exclude 50% of the population without any adverse impact on genetic variation within the population. Although exclusion of dogs with Grade 4 would result in an apparent reduction in the mean BSF score, this reduction is not significant. As NS accounts for 32% of the variation in BSF score, a possible long term strategy to reduce the prevalence of the BS in French bulldogs would seem to be a selection scheme that first excluded dogs with the most severe NS from breeding, gradually moving towards selecting dogs with lower NS grades. According to our findings there is no viable short term solution for reducing the prevalence of BS in the French bulldog population.

The funders had no role in study design, data collection and analysis, decision to publish, or preparation of the manuscript.

**Competing interests:** The authors have declared that no competing interests exist.

## Background

Brachycephalic syndrome (BS) constitutes a major health and welfare problem in several short-nosed breeds as it causes breathing difficulties, heat and exercise intolerance, sleep-disordered breathing, cyanosis and collapse in the affected individuals [1]. In line with the increased popularity of brachycephalic breeds and the demand for hyper-types, the problem seems to be continuously growing. Surgery and medical management can alleviate the symptoms of BS at the individual level, but prophylaxis through breeding is the only available solution at population level. As such, there is an urgent need for breeding efforts to improve the health status of dog breeds affected by BS.

This study aimed to elucidate the possibility of establishing a breeding strategy that could reduce BS in French bulldogs. In order to establish a breeding strategy, two conditions must be met: 1) effective selection criteria must be established, and 2) the proportion of the population deemed phenotypically acceptable for breeding must have sufficient genetic variation. While information on genetic variation can be readily obtained at population level, identification of appropriate selection criteria poses a bigger challenge.

Because health problems related to BS primarily occur due to anatomic abnormalities in the upper airways, quantitative assessment of the anatomical deviation using computer tomography (CT) and laryngoscopy seems to be an obvious way to grade BS. However, no published studies have evaluated whether objective measures of the problematic anatomic abnormalities reflect the degree of functional deterioration. It is the functional impact of the patho-anatomy that affects the daily life of the individual dog and which therefore should be in focus. There are several other reasons for substituting assessment of the anatomical deviations with functional assessment of BS: Firstly CT and laryngoscopy require generalized anesthesia, which poses an increased risk in brachycephalic dogs, and secondly, CT and laryngoscopy are expensive to carry out, which might withhold some breeders from having them performed. Therefore, in the current study, functional assessment is used to identify phenotypic risk factors that may be used as part of an effective breeding strategy.

As BS has evolved in line with intensified breeding for shorter muzzles in brachycephalic breeds, muzzle length has been recognized as the determining factor for BS, despite a lack of evidence relating to the scaling effect [2, 3]. While Packer and others [3] found the risk of BS increased steeply in a non-linear manner concordant with a decrease in the craniofacial ratio (CFR), Liu and others [2] found only a weak association between CFR and the risk of BS in French bulldogs. A short muzzle in itself therefore does not seem to be a major risk factor for BS when evaluated within the range of lengths represented within the French bulldog population. Accordingly, recent studies have identified various other risk factors including sex, body condition score (BCS) and external morphometric features including neck girth (NG) and degree of nostril stenosis (NS) [2, 3].

Although the degree of external NS is not a direct expression of the degree of intranasal stenosis, the inner nasal vestibule is often concomitantly reduced due to a relatively large inner wing of nasal cartilage [4, 5]. While moderate-to-severe stenotic nostrils have been found to increase the mean BS index by only 16% across brachycephalic breeds, it has been shown that the risk of BS is increased by about 20 times in French bulldogs with moderate-to-severely stenotic nares [2, 6].

Exercise testing in the form of a walk test, measured as distance walked during a 6-minute period, has proved effective in capturing differences in physical capacity of dogs with chronic heart failure, mild to moderate pulmonary disease, centronuclear myopathy and in dogs undergoing weight loss [7, 8, 9, 10, 11, 12]. Likewise, it has been shown that the severity of BS in English Bulldogs is negatively correlated with the 6MWT distance and lately, the potential

of exercise testing and laryngeal/tracheal auscultation as a valuable tool for grading of BS has been established [13, 14].

In the present study we use the 6-minute walking test (6MWT), auscultation and dB recordings of upper airway noise to establish Brachycephalic Syndrome Functional scores (BSF scores). We evaluate the correlation between the BSF scores and morphometric and other phenotypic parameters (including parameters that might confound the results), in order to identify potential conformational risk factors applicable as tools in a breeding strategy. To investigate whether it would be possible to implement a selection scheme aimed at reducing the risk of BS at population level without creating a genetic bottleneck, the genetic variation in the total study population was evaluated and compared to the genetic variation in subpopulations selected based on morphometric traits highly correlated with low-level BS.

## Methods

The study was performed at the University of Copenhagen, Department of Veterinary and Animal Science during March and April 2018. The dogs were tested in a quiet, air- and temperature-regulated (21–22˚C) corridor within the department.

The study protocol was approved by the Research Ethics Committee for Science and Health at the University of Copenhagen (Ref 504-0027/18-5000).

### Animal material

A total of 79 French bulldogs were recruited via the social media platform Facebook during March 2018. A public recruitment poster was posted on the first author's personal wall and was shared 269 times by public dog groups, on private walls, and on veterinary pages. A total of 90 dogs unrelated at the parental level were included on first-come, first-served basis. Both FCI-registered and non-registered dogs were accepted for the study. A waiting list was established for all subsequent inquiries. Of the 90 dogs, 11 did not arrive for their appointment or their owners cancelled too late to invite the next on the waiting list. Consequently, the test population comprised 79 dogs. The dogs were 1–5 years of age and had no history of upper airway surgery or any detectable orthopedic, cardiovascular, neurological or pulmonary disease. Intake of medication that could potentially affect the physical capacity and clinical evaluation (e.g. steroids and nonsteroidal anti-inflammatory drugs) within the past 6 weeks was also a reason for exclusion. Informed consent forms were signed by all dog owners before the clinical study was performed.

### Clinical study

The clinical part of the study aimed to (i) register morphometric characteristics that could be correlated to BS, as well as other phenotypic characteristics that could be confounders for the correlations discovered (13 parameters in total), and (ii) use the BSF score as a proxy for degree of BS.

The morphometric and phenotypic characteristics included height (H), neck length (NL), neck girth (NG), the neck-length-to-neck-girth ratio (NLGR), muzzle length (ML), cranial length (CL), craniofacial ratio (CFR), nostril stenosis (NS), BCS and body weight (W). Details on how these parameters were established are provided in Table 1 and Fig 1. Measurements of the skull, neck and height were performed in accordance with [15], while the degree of NS was graded in accordance with [2]. The BCS scoring system used was the 9-point body condition scoring system [16]. The same two investigators scored and agreed on all BCS as well as nostril gradings (intra-observer error was not taken into account), and the latter were subsequently reviewed and confirmed using the rostro-caudal photos taken of nares. One investigator took

**Table 1. Description of measurements of morphometric and phenotypic characteristics.**

| PARAMETER | DESCRIPTION |
|---|---|
| **Nostril senosis (NS)** | |
| Grade 1 | Open nostrils: Nostrils are wide open. |
| Grade 2 | Mildly stenotic nostrils: Slightly narrowed nostrils where the lateral nostril wall does not touch the medial nostril wall. |
| Grade 3 | Moderately stenotic nostrils: The lateral nostril wall touches the medial nostril wall at the dorsal part of the nostril and the nostrils are only open at the bottom. The nostril wings were not able to move dorsolaterally immediately after the exercise test, and there may be nasal flaring. |
| Grade 4 | Severely stenotic nostrils: Nostrils are almost closed. The dog may switch from nasal to oral breathing after stress or very gentle exercise such as playing. |
| **Height (H)** | The linear distance from the ground to the dorsocranial angle of the scapula (withers). Measured when the dog is in an upright, stacked position. |
| **Muzzle length (ML)** | The distance along the skull from the rostral end of the planum nasale to the dorsal plane between the punctae lacrimale of the left and right eye (the stop). |
| **Cranial length (CL)** | The distance along the skull from the stop to the external occipital protuberance. |
| **Neck length (NL)** | The distance along the body from the external occipital protuberance to the dorsal plane of the withers. This is measured along the median plane of the dog. |
| **Neck girth (NG)** | The circumference of the neck at the median distance between the external occipital protuberance and the withers. |
| **Neck girth-to-length ratio (NGLR)** | The ratio of neck girth to neck length: $NGLR = \frac{NG}{NL}$ |
| **Cranio-facial ratio (CFR)** | The ratio of cranial length to muzzle length: $CFR = \frac{CL}{ML}$ |
| **Body Condition Score (BCS)** | [16] |

all the morphometric measurements of the participating dogs to avoid any observer-related systematic error [2]. In addition to the morphometric measurements, the sex, neuter status and age were noted for each participating dog, based on owner information. All morphometric and phenotypic recordings are provided in S1 Table.

The BSF score was used as a proxy for degree of BS. This score is a continuous variable calculated on the basis of 16 assessments of upper airway noise and the presence of respiratory distress at rest during and after a submaximal exercise test and after a 15-minute recovery period. Information relating to 1) respiratory distress 2) audible upper airway noise during auscultation of the a) lungs and b) trachea/larynx, 3) the presence of audible upper airway noise without a stethoscope, and 4) decibel recordings was registered prior to and immediately after a 6-minute walk test (6MWT) and after a 15-minute recovery period. The decibel

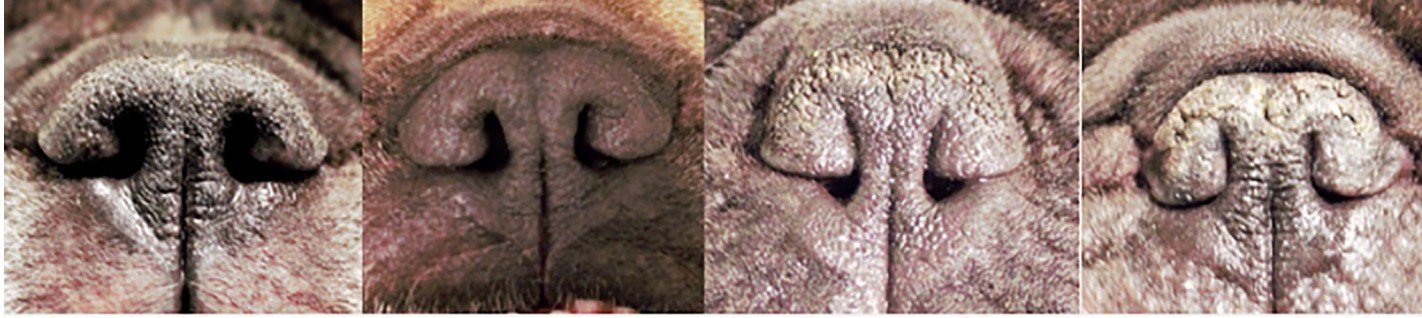

**Fig 1. Representative examples of the four NS grades described in Table 1.** From left to right: Grade 1, Grade 2, Grade 3, Grade 4.

recordings were done using the app "Sound Meter" version 1.1 by Examobile on an iPhone 8, where 0–30 dB was considered baseline (background noise), while 30–40 dB yielded a score of 1, and 40–50 dB a score of 2. The presence of audible upper respiratory noise without a stethoscope was also registered during exercise. The presence of respiratory distress as well as intermittent noise returned a score of 1, while continuous abnormal upper airway noise yielded a score of 2. The scores were summed to generate a total score for each dog ranging from 0–32 (see Table 2 for details on the calculation scheme). All scores for the individual dogs are presented in S2 Table.

## Data processing and statistical analysis

The Spearman Correlation test was used to assess the correlation between non-parametric numeric variables, while the Pearson Correlation test was performed on the variables deemed to follow a normal distribution. The Wilcoxon rank-sum test was used for comparison of the ranks of non-parametric variables with two levels, whereas the Kruskal-Wallis test by Ranks was performed on the nostril grade as it had >2 levels. An unpaired two-tailed t-test without homogeneity of variance was used to compare means for data deemed to follow a normal distribution, and multivariable linear regression analysis was performed with the BSF score as the outcome and including parameters that were both relevant for breeding and significantly correlated with the BSF score as input variables: degree of NS (with three levels: 1–2, 3 and 4, NG (cm), muzzle length (cm), CFR. A backwards stepwise model selection based on Akaike's Information Criterion was used to determine the model with the best fit. The Wilcoxon rank-sum test was used to compare non-parametric parameters. Statistical analyses were performed in R v. 3.5.1 [17] using a significance level of 0.05.

## Genotyping and evaluation of genetic variation

Buccal swabs were collected from all dogs for genetic testing. DNA was extracted from the swabs using the Wizard[R] Genomic DNA Purification Kit (Promega, Madison, USA) for purification of DNA. Microsatellite genotyping was performed using the Canine Genotypes TM

**Table 2. Illustration of the scoring system used to calculate the Brachycephalic Syndrome Functional score (BSF score).**

| | | Score 0 | Score 1 | Score 2 |
|---|---|---|---|---|
| **Pre-exercise** | Presence of respiratory distress | No | Yes | |
| | Presence of audible upper airway noise during auscultation of the lungs | No | Yes, intermittent | Yes, continuously |
| | Presence of audible upper airway noise during auscultation of trachea/larynx | No | Yes, intermittent | Yes, continuously |
| | Presence of upper airway noise audible without stethoscope | No | Yes, intermittent | Yes, continuously |
| | Decibel recordings | 20–30 Hz | 30–40 Hz | 40–50 Hz |
| **During 6MWT** | Presence of upper airway noise audible without stethoscope | No | Yes, intermittent | Yes, continuously |
| **Post-exercise** | Presence of respiratory distress | No | Yes | |
| | Presence of audible upper airway noise during auscultation of the lungs | No | Yes, intermittent | Yes, continuously |
| | Presence of audible upper airway noise during auscultation of trachea/larynx | No | Yes, intermittent | Yes, continuously |
| | Presence of upper airway noise audible without stethoscope | No | Yes, intermittent | Yes, continuously |
| | Decibel recordings | 20–30 Hz | 30–40 Hz | 40–50 Hz |
| **Recovery** | Presence of respiratory distress | No | Yes | |
| | Presence of audible upper airway noise during auscultation of the lungs | No | Yes, intermittent | Yes, continuously |
| | Presence of audible upper airway noise during auscultation of trachea/larynx | No | Yes, intermittent | Yes, continuously |
| | Presence of upper airway noise audible without stethoscope | No | Yes, intermittent | Yes, continuously |
| | Decibel recordings | 20–30 Hz | 30–40 Hz | 40–50 Hz |

panel 1.1, F-860S/L (Finnzymes Diagnostics, Finnzymes Oy, Keilaranta 16 A, 02150 Espoo, Finland) comprising 18 microsatellites distributed on 18 of the 19 canine autosomes. Breed detection was carried out in all the non-FCI registered dogs using the Wisdom Panel[TM] 4.0. Allele frequencies (f(A)), numbers of alleles ($A_A$), effective alleles ($A_{AE}$), observed and expected heterozygosity ($H_O$ and $H_E$, respectively) and inbreeding coefficients (F) were calculated using Microsoft Excel. $A_{AE}$ is defined as the average number of equally frequent alleles (effective alleles) needed to achieve the average level of gene diversity ($A_E = \frac{1}{1-H_E}$). $H_E$ was calculated using the formula: $H_E = 1 - \sum_{i=1}^{n} f(A)_i^2$ where $f(A)_i$ is the frequency of the i[th] of n alleles, while $H_O$ is the heterozygosity present in the population directly derived from the actual geno-type dataset. F is the population inbreeding coefficient ($F = \frac{H_E - H_O}{H_E}$).

For the sake of investigating possibilities for selection, the population was hypothetically divided into three subpopulations. The genetic variation parameters and the BSF scores were compared between the subpopulation of dogs with NS1 and NS2 ($NS_{1-2}$), the subpopulation of dogs with NS1, NS2, and NS3 ($NS_{1-3}$), and the total study population ($NS_{1-4}$). This was achieved by performing an unpaired two-tailed t-test without homogeneity of variance for normally distributed variables and a Wilcoxon rank-sum test for non-parametric variables.

## Results

A total of 79 French bulldogs participated in the study. On the day of testing, 7 dogs were excluded due to: orthopedic (n = 1) or behavior (n = 4) problems, being under one year of age (n = 1) or having an elevated core temperature (n = 1). After breed determination and revision of pedigrees, three dogs were excluded due to being of mixed breeds. Consequently, the total number included in the analyses was 69 dogs.

### Evaluation of the clinical parameters

The BSF scores of the total study population were distributed with a mean value of 14.12 ± 7.22 standard deviations. The scores ranged from 0 to 29 out of 32, with only 3 dogs out of all 69 given a score of 0 (Fig 2).

The correlations between the BSF score and the 14 clinical parameters are presented in Table 3, which shows the mean and median for normally and non-normally distributed parameters, respectively. As can be seen, the population included in the analyses was almost equally distributed in terms of sex at 43% male and 57% female, with a total of 83% of the dogs being intact. The median age was 2.5 years, with the youngest dog tested at exactly 12 months of age and the oldest dog participating one month before turning six years old. The mean weight ± SD was 12.47 ± 2.25 kg, where 62.3% of dogs were overweight (BCS > 5/9).

NLGR, ML, CFR, NG, sex, NS, being overweight and distance covered during 6MWT were all correlated with the BSF score. No significant correlation was found with NL, CL, W, H, age or neuter status. The multivariable regression analysis was carried out using the five signifi-cantly correlated morphometric parameters (CFR, ML, NG, NGLR and NS). The analysis showed that only NG and NS contributed to the model with the best fit (Table 4). The model suggested that NS had the highest impact on the BSF score with 10.59 added to the score if the dog presented with severely stenotic nostrils (Grade 4) instead of open (Grade 1) or mildly ste-notic (Grade 2) nostrils, or an additional 8.45 if the nostrils were moderately stenotic (Grade 3). NS alone could explain 32% of the variation in BSF scores, while NG added 0.50 BSF scores per cm, and could explain 4%.

Based on the multivariable regression analysis, two subpopulations ($NS_{1-2}$ and $NS_{1-3}$) were investigated in order to assess the impact of a potential NS-based breeding strategy. The $NS_{1-2}$

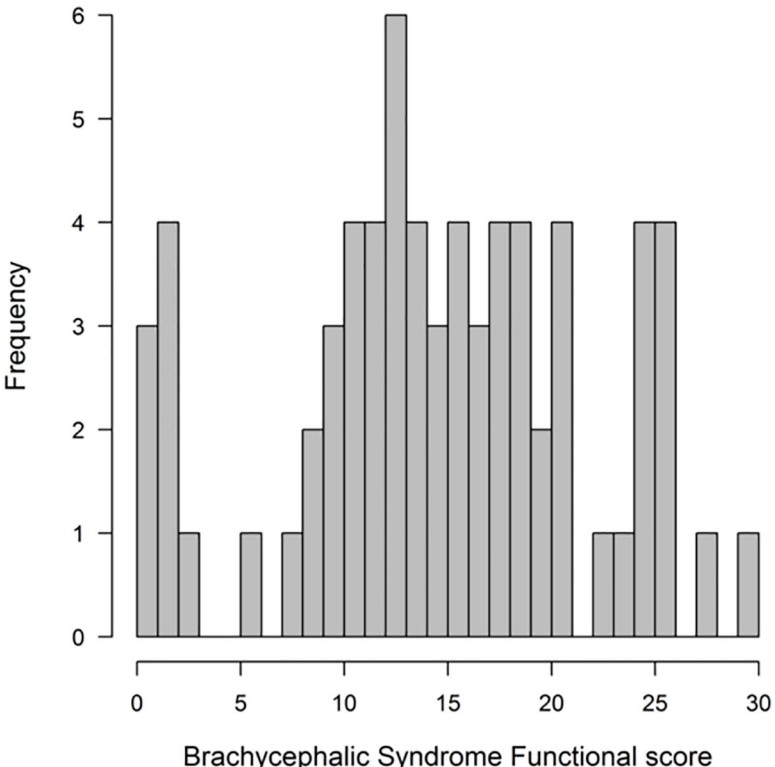

**Fig 2. The distribution of BSF scores within the study population.**

population comprised 13 individuals with Grade 1 and 2 NS, corresponding to 19% of the total study population. The $NS_{1-3}$ population also included individuals with Grade 3 NS, increasing the number of dogs to 35, or 51% of the total study population. The mean BSF score of the $NS_{1-2}$ population was significantly lower than that of the total study population, i.e., $5.62 \pm 6.98$ ($p = 0.0009$), whereas the $NS_{1-3}$ population had a slightly and non-significantly lower mean BSF score of $11.40 \pm 7.07$ ($p = 0.070$).

## Evaluation of genetic variation

All but two samples were successfully genotyped, and a total of 95 alleles were detected across the 18 microsatellite loci of the total study population. The two samples that were not successfully genotyped missed the genotype of only one microsatellite locus each (AHTh260 and FH2848). The remaining 17 microsatellite loci in these samples were included in the study. The genotyping data is presented in S3 Table.

Frequencies of the individual alleles detected in the 18 microsatellites together with the genetic parameters used to evaluate genetic variation in the total population ($NS_{1-4}$) and the two subpopulations ($NS_{1-3}$, and $NS_{1-2}$) are presented in Table 5.

Polymorphism was found in all autosomal loci with a median of five alleles per locus and with the absolute number of alleles per locus ranging from three (AHTk211, REN16C04, REN169D01) to nine (AHT137). In total, 92% of the 95 alleles present in the total study population were represented in the $NS_{1-3}$ population, while 77% were represented in the 13 dogs that made up the $NS_{1-2}$ population. As seen in Table 5, the two subpopulations are comparable to the total population with respect to the number of effective alleles ($A_{AE}$) and the observed ($H_O$) and expected heterozygosity ($H_E$). However, the number of alleles ($A_A$) is significantly

**Table 3. Correlations between BSF score and morphometric and phenotypic parameters in 69 French bulldogs.**

|  | Mean ± SD (min-max) | r (95% CI) | p |
|---|---|---|---|
| **NL** (cm) | 11.17 ± 1.89 (7–17) | -0.15 (-0.37; 0.093) | 0.23[P] |
| **NLGR*** | 0.31 ± 0.050 (0.19–0.44) | -0.31 (-0.51; -0.080) | 0.0094[P] |
| **CL** (cm) | 13.38 ± 1.07 (10–15.5) | 0.17 (-0.065; 0.39) | 0.15[P] |
| **ML*** (cm) | 2.43 ± 0.54 (1.50–4) | -0.26 (-0.47; -0.024) | 0.032[P] |
| **CFR*** | 0.18 ± 0.040 (0.10–0.30) | -0.31 (-0.51; -0.080) | 0.0092[P] |
| **W** (kg) | 12.47 ± 2.25 (7–18.80) | 0.10 (-0.14; 0.33) | 0.39[P] |
| **6MWT distance** | 511 ± 113 (270–752.80) | -0.52 (-0.67; -0.32) | 0.0000056[P] |
|  | Median (min-max) | Þ | p |
| **NG*** (cm) | 36.57 (31–44) | 0.29 | 0.015[s] |
| **H** (cm) | 33.32 (29–39) | -0.16 | 0.20[s] |
| **Age** (months) | 31.87 (12–71) | -0.084 | 0.49[s] |
| **NS*** | 3 (1–4) |  | 0.00008[k] |
| *Grade 1* | 1.4% |  |  |
| *Grade 2* | 17.4% |  |  |
| *Grade 3* | 31.9% |  |  |
| *Grade 4* | 49.3% |  |  |
| **Neuter status** *Neutered* | 17.4% |  | 0.25[w] |
| *Intact* | 82.6% |  |  |
| **Sex*** *Male* | 43.5% |  | 0.017[w] |
| *Female* | 56.5% |  |  |
| **BCS*** *Normal Overweight* | 37.7% |  | 0.037[w] |
|  | 62.3% |  |  |

NL = neck length; NLGR = Neck length to girth ratio; CL = cranial length; ML = muzzle length; CFR = Craniofacial ratio; W = weight; 6MWT = 6-minute walking test;

NG = neck girth; H = height; NS = nostril stenosis; BCS = body condition score

*Variables that are statistically correlated with the BSF score at a significance level of 0.05

r = Pearson Correlation coefficient and associated test (p)

p = Spearman Correlation coefficient and associated test (p)

[k] = Kruskal-Wallis test

[w] = Wilcoxon rank-sum test

lower in the $NS_{1-2}$ population compared to the total population. Nevertheless, since there are no deviations in $H_O$ and $H_E$ in the three populations, the coefficient of inbreeding (F) is close to zero (0.010 ± 0.061, -0.017 ± 0.080 and -0.083 ± 0.18). It should, however, be noted that because of the small sample size in $NS_{1-2}$ (only 13 dogs), the standard deviation for F is high.

**Table 4. Results of the multivariable linear regression analysis of BSF score and morphometric parameters in 69 French bulldogs.**

|  |  | Estimate | SE | P |
|---|---|---|---|---|
| **Intercept** |  | -11.93 | 8.09 | 0.15 |
| **Nostril grade** | *1–2* | 0 | 0 | Reference |
|  | *3* | +8.45 | 2.05 | 0.0001 |
|  | *4* | +10.59 | 1.91 | <0.0001 |
| **Neck girth** (cm) |  | +0.50 | 0.22 | 0.030 |

**Table 5. Genetic parameters.**

| AHT121 | 96 | 98 | 100 | 102 | 104 | 106 | 108 | 112 | | $A_A$ | $A_{AE}$ | $H_O$ | $H_E$ | F |
|---|---|---|---|---|---|---|---|---|---|---|---|---|---|---|
| $NS_{1-4}$ | 0.09 | 0.02 | 0.30 | 0.07 | 0.38 | 0.04 | 0.08 | 0.01 | | 8 | 3.91 | 0.77 | 0.74 | -0.035 |
| $NS_{1-3}$ | 0.13 | 0.03 | 0.29 | 0.11 | 0.36 | 0.04 | 0.04 | | | 7 | 4.05 | 0.74 | 0.75 | 0.018 |
| $NS_{1-2}$ | 0.23 | 0.04 | 0.27 | 0.08 | 0.35 | 0.04 | | | | 6 | 3.88 | 0.77 | 0.74 | -0.038 |

| AHT137 | 131 | 133 | 135 | 137 | 141 | 147 | 149 | 151 | 153 | $A_A$ | $A_{AE}$ | $H_O$ | $H_E$ | F |
|---|---|---|---|---|---|---|---|---|---|---|---|---|---|---|
| $NS_{1-4}$ | 0.10 | 0.34 | 0.28 | 0.07 | 0.01 | 0.07 | 0.01 | 0.10 | 0.02 | 9 | 4.46 | 0.71 | 0.78 | 0.085 |
| $NS_{1-3}$ | 0.09 | 0.33 | 0.27 | 0.06 | | 0.09 | 0.01 | 0.14 | 0.01 | 8 | 4.52 | 0.71 | 0.78 | 0.088 |
| $NS_{1-2}$ | 0.04 | 0.31 | 0.27 | 0.08 | | 0.12 | | 0.15 | 0.04 | 7 | 4.64 | 0.69 | 0.78 | 0.121 |

| AHTh171 | 219 | 225 | 231 | 233 | 235 | $A_A$ | $A_{AE}$ | $H_O$ | $H_E$ | F |
|---|---|---|---|---|---|---|---|---|---|---|
| $NS_{1-4}$ | 0.31 | 0.49 | 0.01 | 0.19 | 0.01 | 5 | 2.68 | 0.65 | 0.63 | -0.036 |
| $NS_{1-3}$ | 0.36 | 0.43 | 0.01 | 0.20 | | 4 | 2.82 | 0.66 | 0.65 | -0.023 |
| $NS_{1-2}$ | 0.31 | 0.46 | | 0.23 | | 3 | 2.77 | 0.62 | 0.64 | 0.030 |

| AHTh260 | 238 | 244 | 246 | 248 | 250 | 252 | $A_A$ | $A_{AE}$ | $H_O$ | $H_E$ | F |
|---|---|---|---|---|---|---|---|---|---|---|---|
| $NS_{1-4}$ | 0.63 | 0.01 | 0.05 | 0.15 | 0.10 | 0.06 | 6 | 2.30 | 0.57 | 0.56 | -0.0099 |
| $NS_{1-3}$ | 0.69 | | 0.07 | 0.10 | 0.13 | 0.01 | 5 | 1.97 | 0.49 | 0.49 | 0.0041 |
| $NS_{1-2}$ | 0.69 | | 0.04 | 0.12 | 0.15 | | 4 | 1.94 | 0.46 | 0.49 | 0.052 |

| AHTk211 | 87 | 91 | 95 | $A_A$ | $A_{AE}$ | $H_O$ | $H_E$ | F |
|---|---|---|---|---|---|---|---|---|
| $NS_{1-4}$ | 0.62 | 0.01 | 0.38 | 3 | 1.89 | 0.45 | 0.47 | 0.045 |
| $NS_{1-3}$ | 0.63 | 0.01 | 0.36 | 3 | 1.90 | 0.46 | 0.47 | 0.028 |
| $NS_{1-2}$ | 0.69 | 0.04 | 0.27 | 3 | 1.82 | 0.46 | 0.45 | -0.024 |

| AHTk253 | 286 | 288 | 290 | 292 | $A_A$ | $A_{AE}$ | $H_O$ | $H_E$ | F |
|---|---|---|---|---|---|---|---|---|---|
| $NS_{1-4}$ | 0.22 | 0.14 | 0.19 | 0.44 | 4 | 3.36 | 0.67 | 0.70 | 0.046 |
| $NS_{1-3}$ | 0.20 | 0.14 | 0.26 | 0.40 | 4 | 3.48 | 0.77 | 0.71 | -0.080 |
| $NS_{1-2}$ | 0.15 | 0.19 | 0.35 | 0.31 | 4 | 3.61 | 0.77 | 0.72 | -0.065 |

| CXX279 | 116 | 118 | 120 | 124 | 126 | 131 | $A_A$ | $A_{AE}$ | $H_O$ | $H_E$ | F |
|---|---|---|---|---|---|---|---|---|---|---|---|
| $NS_{1-4}$ | 0.01 | 0.04 | 0.17 | 0.54 | 0.22 | 0.02 | 6 | 2.70 | 0.62 | 0.63 | 0.014 |
| $NS_{1-3}$ | 0.01 | 0.01 | 0.20 | 0.53 | 0.21 | 0.03 | 6 | 2.73 | 0.63 | 0.63 | 0.0062 |
| $NS_{1-2}$ | | | 0.23 | 0.42 | 0.27 | 0.08 | 4 | 3.24 | 0.77 | 0.69 | -0.11 |

| FH2054 | 148 | 152 | 156 | 160 | 164 | 168 | 172 | 176 | $A_A$ | $A_{AE}$ | $H_O$ | $H_E$ | F |
|---|---|---|---|---|---|---|---|---|---|---|---|---|---|
| $NS_{1-4}$ | 0.15 | 0.05 | 0.45 | 0.02 | 0.02 | 0.04 | 0.26 | 0.01 | 8 | 3.36 | 0.65 | 0.70 | 0.075 |
| $NS_{1-3}$ | 0.19 | 0.09 | 0.39 | 0.03 | 0.04 | 0.04 | 0.23 | | 7 | 3.95 | 0.77 | 0.75 | -0.031 |
| $NS_{1-2}$ | 0.19 | 0.12 | 0.35 | | 0.08 | 0.04 | 0.23 | | 6 | 4.28 | 0.77 | 0.77 | -0.0051 |

| FH2848 | 230 | 236 | 238 | 240 | 244 | $A_A$ | $A_{AE}$ | $H_O$ | $H_E$ | F |
|---|---|---|---|---|---|---|---|---|---|---|
| $NS_{1-4}$ | 0.02 | 0.04 | 0.26 | 0.60 | 0.08 | 5 | 2.29 | 0.56 | 0.56 | 0.0071 |
| $NS_{1-3}$ | 0.03 | 0.07 | 0.29 | 0.54 | 0.07 | 5 | 2.59 | 0.69 | 0.61 | -0.12 |
| $NS_{1-2}$ | 0.04 | 0.08 | 0.31 | 0.46 | 0.12 | 5 | 3.029 | 0.77 | 0.67 | -0.15 |

| INRA21 | 95 | 97 | 99 | 101 | 105 | $A_A$ | $A_{AE}$ | $H_O$ | $H_E$ | F |
|---|---|---|---|---|---|---|---|---|---|---|
| $NS_{1-4}$ | 0.83 | 0.05 | 0.07 | 0.01 | 0.04 | 5 | 1.43 | 0.33 | 0.30 | -0.093 |
| $NS_{1-3}$ | 0.86 | 0.04 | 0.04 | | 0.06 | 4 | 1.34 | 0.29 | 0.25 | -0.14 |
| $NS_{1-2}$ | 0.88 | 0.04 | 0.04 | | 0.04 | 4 | 1.28 | 0.31 | 0.22 | -0.40 |

| INU005 | 110 | 122 | 124 | 126 | 130 | $A_A$ | $A_{AE}$ | $H_O$ | $H_E$ | F |
|---|---|---|---|---|---|---|---|---|---|---|
| $NS_{1-4}$ | 0.04 | 0.01 | 0.38 | 0.30 | 0.25 | 5 | 3.35 | 0.67 | 0.70 | 0.045 |
| $NS_{1-3}$ | 0.03 | 0.01 | 0.49 | 0.29 | 0.19 | 5 | 2.77 | 0.69 | 0.64 | -0.080 |
| $NS_{1-2}$ | 0.08 | | 0.38 | 0.31 | 0.23 | 4 | 3.34 | 0.85 | 0.70 | -0.21 |

| INU030 | 144 | 150 | 152 | 154 | 156 | $A_A$ | $A_{AE}$ | $H_O$ | $H_E$ | F |
|---|---|---|---|---|---|---|---|---|---|---|
| $NS_{1-4}$ | 0.01 | 0.70 | 0.28 | 0.01 | 0.01 | 5 | 1.76 | 0.43 | 0.43 | 0.0030 |

(Continued)

**Table 5.** (Continued)

| Microsatellites | | | | | | | $A_A$ | $A_{AE}$ | $H_O$ | $H_E$ | F |
|---|---|---|---|---|---|---|---|---|---|---|---|
| NS$_{1-3}$ | 0.01 | 0.69 | 0.29 | | 0.01 | | 4 | 1.78 | 0.49 | 0.44 | -0.11 |
| NS$_{1-2}$ | | 0.85 | 0.12 | | 0.04 | | 3 | 1.35 | 0.31 | 0.26 | -0.19 |
| **INU055** | **208** | **210** | **212** | **218** | **220** | | | | | | |
| NS$_{1-4}$ | 0.01 | 0.42 | 0.01 | 0.55 | 0.01 | | 5 | 2.087 | 0.52 | 0.52 | 0.0015 |
| NS$_{1-3}$ | 0.01 | 0.44 | 0.01 | 0.51 | 0.01 | | 5 | 2.20 | 0.51 | 0.55 | 0.066 |
| NS$_{1-2}$ | 0.04 | 0.42 | | 0.54 | | | 3 | 2.13 | 0.77 | 0.53 | -0.45 |
| **REN16C04** | **202** | **204** | **206** | | | | | | | | |
| NS$_{1-4}$ | 0.64 | 0.13 | 0.22 | | | | 3 | 2.11 | 0.46 | 0.53 | 0.12 |
| NS$_{1-3}$ | 0.66 | 0.10 | 0.24 | | | | 3 | 1.99 | 0.49 | 0.50 | 0.014 |
| NS$_{1-2}$ | 0.54 | 0.15 | 0.31 | | | | 3 | 2.44 | 0.54 | 0.59 | 0.084 |
| **REN169D01** | **212** | **216** | **218** | | | | | | | | |
| .NS$_{1-4}$ | 0.32 | 0.36 | 0.33 | | | | 3 | 2.93 | 0.68 | 0.66 | -0.032 |
| NS$_{1-3}$ | 0.37 | 0.27 | 0.36 | | | | 3 | 2.95 | 0.63 | 0.66 | 0.046 |
| NS$_{1-2}$ | 0.54 | 0.27 | 0.19 | | | | 3 | 2.50 | 0.69 | 0.60 | -0.151 |
| **REN169O18** | **156** | **164** | **166** | **168** | **170** | **172** | | | | | |
| NS$_{1-4}$ | 0.09 | 0.01 | 0.06 | 0.31 | 0.49 | 0.04 | 6 | 2.86 | 0.71 | 0.65 | -0.092 |
| NS$_{1-3}$ | 0.06 | 0.01 | 0.03 | 0.40 | 0.46 | 0.04 | 6 | 2.65 | 0.57 | 0.62 | 0.084 |
| NS$_{1-2}$ | 0.08 | | | 0.38 | 0.46 | 0.08 | 4 | 2.71 | 0.54 | 0.63 | 0.15 |
| **REN247M23** | **266** | **268** | **270** | **272** | | | | | | | |
| NS$_{1-4}$ | 0.01 | **0.51** | 0.10 | 0.37 | | | 4 | 2.46 | 0.54 | 0.59 | 0.089 |
| NS$_{1-3}$ | | 0.43 | 0.16 | 0.41 | | | 3 | 2.64 | 0.57 | 0.62 | 0.083 |
| NS$_{1-2}$ | | 0.46 | 0.04 | 0.50 | | | 3 | 2.16 | 0.46 | 0.54 | 0.14 |
| **REN54P11** | **222** | **226** | **228** | **234** | **236** | | | | | | |
| NS$_{1-4}$ | 0.07 | 0.13 | 0.61 | 0.04 | 0.16 | | 5 | 2.37 | 0.61 | 0.58 | -0.054 |
| NS$_{1-3}$ | 0.10 | 0.10 | 0.56 | 0.03 | 0.21 | | 5 | 2.64 | 0.71 | 0.62 | -0.14 |
| NS$_{1-2}$ | 0.12 | 0.04 | 0.50 | | 0.35 | | 4 | 2.57 | 0.77 | 0.61 | -0.26 |

NS$_{1-4}$ = total population; NS$_{1-3}$ = dogs with nostril score 1, 2 and 3; NS$_{1-2}$ = dogs with nostril score 1 and 2; $A_A$ = number of alleles; $A_{AE}$ = number of effective alleles; $H_O$ = observed heterozygosity; $H_E$ = expected heterozygosity; F = inbreeding coefficient.

## Discussion

Due to the severe health and welfare problems experienced by brachycephalic dogs [1, 5] with exaggerated phenotypes, several studies have been performed to evaluate the severity of BS based on exercise tests [13, 18]. In the present study we have introduced the calculation of BSF scores, on a continuous scale, that rely on recordings of upper airway noise and decibel recordings before, during, and after exercise to assess the degree of BS. Some of the correlations between BSF score and morphometric and phenotypic variables (Table 3) are substantial, e.g. walking distance and neck length to girth ratio, whereas others are minor e.g. weight and cranial length. Generally the correlations make sense, because weight in itself would rarely be expected to have a major impact as it is an unspecific parameter, whereas walking distance is a patho-physiologic adverse outcome. Although BSF scores cannot be considered as a definitive diagnostic tool they have been shown to reflect the severity of BS. To lend support to the functional characterization, efforts have also been made to identify the phenotypic characteristics that present the highest risk, to aid in selection for healthier conformation. Since the acute and chronic respiratory distress inherent in dogs with BS appears to increase in line with a decreasing muzzle length, it can be assumed that ML and CFR are risk factors. There is, however,

**Table 6. Summary of genetic data in the total population and subpopulations.**

|  | Total study population | $p$ | NS$_{1-3}$ | $p$ | NS$_{1-2}$ |
|---|---|---|---|---|---|
| **A$_A$/locus** *Median (min-max)* | 5 (3–9) | 0.019$^{w*}$ | 5 (3–8) | 0.44$^w$ | 4 (3–7) |
| **A$_{AE}$/locus** *Mean ±SD (min-max)* | 2.68 ± 0.78 (1.43–4.46) | 0.79$^t$ | 2.72 ± 0.81 (1.34–4.52) | 0.89$^t$ | 2.76 ± 0.94 (1.28–4.64) |
| **H$_O$** *Mean ±SD (min-max)* | 0.59 ± 0.12 (0.33–0.77) | 0.42$^t$ | 0.60 ± 0.13 (0.29–0.77) | 0.72$^t$ | 0.63 ± 0.17 (0.31–0.85) |
| **H$_E$** *Mean ±SD (min-max)* | 0.60 ± 0.12 (0.30–0.77) | 0.90$^t$ | 0.60 ± 0.13 (0.25–0.78) | 0.99$^t$ | 0.59 ± 0.16 (0.22–0.78) |
| **F** *Mean ±SD (min-max)* | 0.010 ± 0.061 (-0.093–0.12) | 0.046$^{t*}$ | -0.017 ± 0.080 (-0.14–0.88) | 0.26$^t$ | -0.083 ± 0.18 (-0.45–0.14) |

A$_A$ = number of alleles; A$_{AE}$ = number of effective alleles; H$_O$ = observed heterozygosity; H$_E$ = expected heterozygosity; F = Inbreeding coefficient; $^w$ = Wilcoxon rank sum test, $^t$ = t-test; $p$ = the p-value for the test of difference in means in the two subpopulations NS$_{1-2}$ and NS$_{1-3}$, compared to NS$_{1-4}$. Variables that are significantly different when compared to NS$_{1-4}$ at a significance level of 0.05 are marked with $^*$.

contradicting evidence about the impact of these conformational features on the development of the syndrome. In agreement with another study on French bulldogs [2], we found no significant effect of ML, whereas NS explained 32% of the variation in BSF scores. Our study also corroborates several other studies that indicate NG and BCS are further risk factors for BS [2, 3, 6]. NG and BCS explained 4% and 8% of the variation in BSF scores in our French bulldog population, respectively.

As a result of selection for specific conformational and/or performance traits, all modern breed dogs exhibit reduced genetic diversity. It is therefore of utmost importance that we take genetic variation into consideration when contemplating selection schemes. The average number of alleles per locus (A$_A$) and the average effective alleles per locus (A$_{AE}$) identified in the French bulldogs was similar to other pure breeds that have been monitored for diversity [19], as well as English bulldogs [20]. However, the genetic diversity seems to be higher in French bulldogs than English bulldogs since only 2 of the 18 loci (11%) were approaching fixation in our population (a single allele with frequency >70), while the corresponding number was 7 out of 33 loci (21%) in English bulldogs. In addition, while most of the loci (58%) had one or two alleles that dominated in the English bulldog population, the corresponding figure was 44% in the French bulldog population. The only significant difference between the total population and the two subpopulations (NS$_{1-3}$ and NS$_{1-2}$) was the reduction in the average number of alleles in the NS$_{1-2}$ subpopulation. This is not remarkable since this subpopulation was made up of only 13 dogs. Due to sampling errors in this small subpopulation, the genetic parameters are not very reliable, which is most notably reflected in the very high mean SD for F (see Table 6).

Our study unambiguously shows that the NS score has a large impact on the functional ability of French bulldogs. This single parameter results in an additional 10.59 and 8.45 score points in dogs with open or mildly stenotic nares, respectively. It follows that dogs with open nares are more likely to lead a healthier life compared to dogs with stenotic nares. Hence, selecting only dogs with open nares (NS$_{1-2}$) for breeding would seem to be an obvious means to improve functional ability and, in turn, health and welfare at the population level, and would in fact comply with the FCI standard for the breed, which states that the nose should 'allow normal nasal breathing'[21].

Nevertheless, since this would exclude 81% of the population from breeding, this is obviously not a prudent solution. The average level of genetic diversity in the French bulldog population is comparable to many other pure dog breeds, implying that selection should be performed cautiously while also taking other health aspects into consideration. The breed has a high prevalence of, for instance, dystocia [22] and thoracic vertebral malformations (hemivertebrae) [23]–both of which are severe problems that must be considered during selection. A

lenient way to ensure that the prevalence of BS is reduced among French bulldogs would be to avoid using dogs with severely stenotic nares for breeding. This would lead to progress, albeit at a very slow pace since dogs with moderately stenotic nares would have a negative impact on the progress. However, slow progress must be accepted to avoid a devastating reduction in genetic diversity within the breed.

In spite of the limited number of animals included in the present study, we have confirmed that NS has a considerable impact on BS evaluated based on the BSF score. The main short-coming of this and other studies where the BSF score has been used to reflect the severity of BS is that the BSF score is not a definitive diagnostic evaluation. One possible way to validate the diagnostic value of the score is to compare the score with results from laryngoscopic evaluation of the problematic anatomic abnormalities combined with diagnostic imaging such as CT. This would require general anesthesia and was not performed due to the considerable risk posed to brachycephalic dogs, as mentioned previously. Nevertheless, lending support to the diagnostic value of the BSF score, the study by Riggs and co-workers [14] show that the association between subjectively determined functional grades (BSF score) and objectively determined BS indices, as determined by whole body barometric plethysmography, is high.

## Supporting information

**S1 Table. Morphometric and phenotypic registrations.**
(XLSX)

**S2 Table. Registration of upper airway noise and presence of respiratory distress at rest, during and after a submaximal exercise test and after a 15-minute recovery period.**
(XLSX)

**S3 Table. Microsatellite genotyping results.**
(XLSX)

## Acknowledgments

The authors wish to thank Charlotte Bjørner Larsen for excellent technical assistant.

## Author Contributions

**Conceptualization:** Merete Fredholm.

**Formal analysis:** Eva-Marie Ravn-Mølby, Line Sindahl, Søren Saxmose Nielsen.

**Funding acquisition:** Merete Fredholm.

**Investigation:** Eva-Marie Ravn-Mølby, Line Sindahl.

**Methodology:** Merete Fredholm.

**Supervision:** Camilla S. Bruun, Peter Sandøe, Merete Fredholm.

**Validation:** Søren Saxmose Nielsen.

**Writing – original draft:** Eva-Marie Ravn-Mølby, Line Sindahl, Merete Fredholm.

**Writing – review & editing:** Eva-Marie Ravn-Mølby, Line Sindahl, Søren Saxmose Nielsen, Camilla S. Bruun, Peter Sandøe, Merete Fredholm.

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
