## [Decision Letter · Decision Letter 0]

12 Aug 2019

PONE-D-19-19251

Breeding French bulldogs that are able to breathe - possible but a long way to go

PLOS ONE

Dear Dr. Fredholm,

Thank you for submitting your manuscript to PLOS ONE. After careful consideration, we feel that it has merit but does not fully meet PLOS ONE’s publication criteria as it currently stands. Therefore, we invite you to submit a revised version of the manuscript that addresses the points raised during the review process.

We would appreciate receiving your revised manuscript by Sep 26 2019 11:59PM. To enhance the reproducibility of your results, we recommend that if applicable you deposit your laboratory protocols in protocols.io, where a protocol can be assigned its own identifier (DOI) such that it can be cited independently in the future. For instructions see: http://journals.plos.org/plosone/s/submission-guidelines#loc-laboratory-protocols

We look forward to receiving your revised manuscript.

Kind regards,

Juan J Loor

Academic Editor

PLOS ONE

2. Thank you for stating the following in the Acknowledgments Section of your manuscript: "We also wish to thank the Danish Kennel Club for financial support to the project."

 "No"

Reviewers' comments:

Reviewer's Responses to Questions

**Comments to the Author**

1. Is the manuscript technically sound, and do the data support the conclusions?

Reviewer #1: Yes

Reviewer #2: No

2. Has the statistical analysis been performed appropriately and rigorously? 

Reviewer #1: Yes

Reviewer #2: Yes

3. Have the authors made all data underlying the findings in their manuscript fully available?

Reviewer #1: Yes

Reviewer #2: Yes

4. Is the manuscript presented in an intelligible fashion and written in standard English?

Reviewer #1: Yes

Reviewer #2: Yes

5. Review Comments to the Author

Reviewer #1: Abstract: the authors are definite that NS is the only way to improve BOAS by breeding selection- what about functional grading or genetic testing to gain estimated breeding values?

Also are you sure that NS is a single gene trait- ie progeny from NS1/2 will not be NS3/4? if so please cite the references/ evidence.

54. Not necessarily, anatomical lesions do not always equate to disease severity, particularly as BOAS is a dynamic disease and static pictures may not reveal the extent of disease. Are you talking about direct visualisation? or rhinoscopy? or CT? and if so what criteria are you using to grade disease severity from anatomical lesions?

61 BS increased steeply between breeds...

116 Intra-observer error checked? tricky measurements....

125 Decibel recording- was this based on previous data/ pilot studies or published studies?

What about open mouth vs nasal breathing? ie dogs with severe nasal obstruction may mouth breathe throughout exercise and sound better than dogs with mild/moderate nasal obstruction that nasal breathe throughout exercise- how do you correct for this?

129 I would argue that respiratory distress plus intermittent laryngeal noise is a more severe presentation than a continuous nasal stertor.

This is an interesting study (I need a second reviewer to check the genetics) but the functional grading is not without flaws that should be discussed.

Reviewer #2: The submission entitled “Breeding French bulldogs that able to breathe – possible but a long way to go” addresses and an incredibly important topic and the authors present some valuable findings however there are many concerns with the manuscript in its current form. The authors should be lauded for their efforts!

Major Concerns:

The title is inappropriate. French bulldogs are able to breathe in their current physiological form. The title implies they cannot which is a misnomer. Their work addresses the view that it would be possible to breed with intention for improved nares structure plus other attributes to reduce the incidence of brachycephalic syndrome.

The presentation in its current form is disingenuous. The authors are really developing a model to characterize overall brachycephalic syndrome (BS) and the contributory factors. The authors then use that descriptive rubric to model the effects on the French bulldog population if selection using that rubric were employed. Several main concerns center on this. The authors present the brachycephalic syndrome functional (BSF) score as approved methodology, yet this manuscript is just describing the BSF. Importantly the BSF has not been validated to demonstrate that the BSF score accurately predicts a dogs phenotype in a replicate population.

The authors should be commended for assessing whether implementation of the BSF score in breeding would create genetic bottlenecks that would be harmful for the breed population. However, the abstract implies, as written, that the breeding selection was done using the BSF when in fact it was a hypothetical modeling of selection not actual implementation. This distinction must be clearly stated in the abstract, in the methods, in the results, and in the discussion.

Furthermore, the authors come to the conclusion that genetic variability within the population would be limited based upon data from 18 microsatellites. This is a limitation of the study and needs to be discussed. It is also curious why the authors only used those 18 microsatellites—yes they are on a panel that can be used but the rationale as to why these 18 are sufficient to reflect breed population diversity must be discussed. And how these msats are distributed across the genome.

In the discussion, the authors state “Our study unambiguously shows that the NS score has a large impact on the health and welfare of French bulldogs”. Their data demonstrate that the BSF score based upon the measurements taken on a sample of French bulldogs correlated with the measurements that were used to form the BSF score. The BSF score includes physiological parameters of heart rate, body temperature, etc—these features are key to include in the main text. To find the components, they are listed in the supplemental material. The manuscript would be much more powerful if the elements of the BSF were included. The authors need to assess welfare and health using their BSF score and the NS on a validation population before making such a statement. If the authors did this and demonstrated that functional welfare and health parameters are predicted by their BSF score that would be a huge advancement in the field and for the brachycephalic dogs. Unfortunately, that is not what is presented in the current manuscript.

Line 18: should be “those” not these

Line 28 should be no or only mild NS

Line 31-32 “Although it results in apparent reduction…” unclear antecedent. To what is “it” referring?

Line 35-36 it is feasible but not prudent to undertake removing all dogs with the severest form of BS.

Lines 73-76 need to be rewritten to reflect that this is a model development paper and that the BSF scoring system does not currently exist. OR the authors must provide a reference for the validation of this tool.

Line 105, Line 119. See comment for lines 73-75. The scoring system does not yet exist and the authors are modeling such a scoring system in this manuscript. That needs to be more explicitly stated and better defined. If however, that BSF currently exists and is in use, the authors need to reference that and demonstrate its validity.

Line 125 reference 9 is a human 6MWT reference based upon bipedal locomotion. The authors will need to mention why this is a valid application for a quadruped dog.

Line 185. The dogs have biological sex (male or female) not gender, which is a social, construct. Change gender to sex throughout the paper.

Line 257 needs referencing.

Line 260 needs referencing.

6. PLOS authors have the option to publish the peer review history of their article (what does this mean?). If published, this will include your full peer review and any attached files.

Reviewer #1: No

Reviewer #2: No

---

## [Author Response · Author response to Decision Letter 0]

20 Sep 2019

Comments to the Author

Reviewer #1: Abstract: the authors are definite that NS is the only way to improve BOAS by breeding selection- what about functional grading or genetic testing to gain estimated breeding values?

We agree with the reviewer and have therefore modified the statement in the abstract to indicate that selection based on NS score is not the only possible selection scheme (l. 39 – l. 40)

Also are you sure that NS is a single gene trait- ie progeny from NS1/2 will not be NS3/4? if so please cite the references/ evidence.

We concede that NS is not a single gene trait - it is a trait for which the heritability has not been established. However, since the NS accounts for 32% of the variation in the BSF score it appears to be a good proxy for the health status. 

54. Not necessarily, anatomical lesions do not always equate to disease severity, particularly as BOAS is a dynamic disease and static pictures may not reveal the extent of disease. Are you talking about direct visualisation? or rhinoscopy? or CT? and if so what criteria are you using to grade disease severity from anatomical lesions?

We agree with the reviewer that BOAS/BS is a dynamic disease. We have explained this in more detail in l. 60 – l. 67. Although there is no ‘golden standard’ for assessing the degree of BOAS/BS it is generally accepted that exercise testing and laryngeal and tracheal auscultation can be used for grading of BS. In l. 85 - l. 89 we have explained this in more detail and included relevant references.

61 BS increased steeply between breeds...

We are not sure what the reviewer is alluding to here. However, BS cannot “increase” between breeds, because “breed” is a categorical variable. 

116 Intra-observer error checked? tricky measurements....

We agree with the reviewer: The parameters with the highest likelihood of intra-observer error were NS and BCS. We have stated that intra-observer error was not taken into account (l. 127 - 128)

125 Decibel recording- was this based on previous data/ pilot studies or published studies?

What about open mouth vs nasal breathing? ie dogs with severe nasal obstruction may mouth breathe throughout exercise and sound better than dogs with mild/moderate nasal obstruction that nasal breathe throughout exercise- how do you correct for this?

Decibel recordings were included in an attempt to strengthen the objectivity of the evaluation of ‘upper airway noice’. To our knowledge this has not been done before. The recordings were made before exercise, right after exercise, and after a 15 minute resting period without taking NS or other factors into consideration. As can be seen in S2 Table (columns J, S and AB), there is a very high correlation with respect to the dogs in which these recordings were positive across recording time points. Decibel recordings are one of several observations to evaluate the degree of BS. In combination with other assessments of airway noise and registrations of respiratory distress it contributes information to the combined BSF score. 

129 I would argue that respiratory distress plus intermittent laryngeal noise is a more severe presentation than a continuous nasal stertor.

This is an interesting study (I need a second reviewer to check the genetics) but the functional grading is not without flaws that should be discussed.

Thank you for the kind words and the careful review of our study. We have explained the BSF scores were established, on a continuous scale, based on recordings of upper airway noise and decibel recordings in the discussion, and acknowledged that the grading is not a definitive diagnostic tool (l. 276– l. 281). 

Reviewer #2: The submission entitled “Breeding French bulldogs that able to breathe – possible but a long way to go” addresses and an incredibly important topic and the authors present some valuable findings however there are many concerns with the manuscript in its current form. The authors should be lauded for their efforts!

Thank you for the kind words and for the careful review of our paper.

Major Concerns:

The title is inappropriate. French bulldogs are able to breathe in their current physiological form. The title implies they cannot which is a misnomer. Their work addresses the view that it would be possible to breed with intention for improved nares structure plus other attributes to reduce the incidence of brachycephalic syndrome.

We agree that the title was misleading. The title is now: Breeding French bulldogs with unobstructed breathing - possible but a long way to go

The presentation in its current form is disingenuous. The authors are really developing a model to characterize overall brachycephalic syndrome (BS) and the contributory factors. The authors then use that descriptive rubric to model the effects on the French bulldog population if selection using that rubric were employed. Several main concerns center on this. The authors present the brachycephalic syndrome functional (BSF) score as approved methodology, yet this manuscript is just describing the BSF. Importantly the BSF has not been validated to demonstrate that the BSF score accurately predicts a dogs phenotype in a replicate population.

We agree with the reviewer and have acknowledged this by changing the wording in the final part of the discussion (l. 324 – l. 334) explaining that validation is needed.

The authors should be commended for assessing whether implementation of the BSF score in breeding would create genetic bottlenecks that would be harmful for the breed population. However, the abstract implies, as written, that the breeding selection was done using the BSF when in fact it was a hypothetical modeling of selection not actual implementation. This distinction must be clearly stated in the abstract, in the methods, in the results, and in the discussion.

We concede that the way we have described the subpopulations could lead to misunderstandings. We have now made it clear that the subpopulations are hypothetical populations (l. 30-33 and l. 180 - 181). 

Furthermore, the authors come to the conclusion that genetic variability within the population would be limited based upon data from 18 microsatellites. This is a limitation of the study and needs to be discussed. It is also curious why the authors only used those 18 microsatellites—yes they are on a panel that can be used but the rationale as to why these 18 are sufficient to reflect breed population diversity must be discussed. And how these msats are distributed across the genome.

We have included information on the distribution of the 18 microsatellites genotyped in this study (distributed on 18 of the 19 canine autosomes) (see l. 170 -171). We concede that it is a low number of markers, still microsatellites are highly polymorphic, neutral to selection and an equivalent number of markers have been used (successfully) in other population studies in dogs (https://doi.org/10.1371/journal.pone.0221418

https://doi.org/10.1017/S1751731117003573).

In the discussion, the authors state “Our study unambiguously shows that the NS score has a large impact on the health and welfare of French bulldogs”. Their data demonstrate that the BSF score based upon the measurements taken on a sample of French bulldogs correlated with the measurements that were used to form the BSF score. The BSF score includes physiological parameters of heart rate, body temperature, etc—these features are key to include in the main text. To find the components, they are listed in the supplemental material. The manuscript would be much more powerful if the elements of the BSF were included. 

Thank you for pointing this out to us. We have now included S2 Table as Table 2 in the manuscript in order to ensure that the readers can see immediately how the BSF scores are established.

The authors need to assess welfare and health using their BSF score and the NS on a validation population before making such a statement. If the authors did this and demonstrated that functional welfare and health parameters are predicted by their BSF score that would be a huge advancement in the field and for the brachycephalic dogs. Unfortunately, that is not what is presented in the current manuscript.

We accept this point which was also made by the other referee. We have acknowledged this by changing the wording in the final part of the discussion (l. 324 – l. 334) explaining that validation is needed.

Line 18: should be “those” not these

Has been corrected (l. 20)

Line 28 should be no or only mild NS

Has been corrected (l. 31)

Line 31-32 Although it results in apparent reduction…” unclear antecedent. To what is “it” referring?

Has been explained (‘it’ refers to exclusion of dogs with Grade 4) (l. 35)

Line 35-36 it is feasible but not prudent to undertake removing all dogs with the severest form of BS.

Has been corrected (‘feasible’ substituted with ‘prudent’) – l. 39.

Lines 73-76 need to be rewritten to reflect that this is a model development paper and that the BSF scoring system does not currently exist. OR the authors must provide a reference for the validation of this tool.

Relevant references have been included. See line 85.

Line 105, Line 119. See comment for lines 73-75. The scoring system does not yet exist and the authors are modeling such a scoring system in this manuscript. That needs to be more explicitly stated and better defined. If however, that BSF currently exists and is in use, the authors need to reference that and demonstrate its validity.

See our response to the previous point.

Line 125 reference 9 is a human 6MWT reference based upon bipedal locomotion. The authors will need to mention why this is a valid application for a quadruped dog.

Thank you for pointing this out. The mentioned reference has been substituted with a more relevant reference.

Line 185. The dogs have biological sex (male or female) not gender, which is a social, construct. Change gender to sex throughout the paper.

Has been corrected

Line 257 needs referencing.

Reference 1 and 2 have been included (l. 275)

Line 260 needs referencing.

Reference 9 and 10 have been included (l. 277)

---

## [Decision Letter · Decision Letter 1]

21 Oct 2019

PONE-D-19-19251R1

Breeding French bulldogs with unobstructed breathing - possible but a long way to go

PLOS ONE

Dear Dr. Fredholm,

Thank you for submitting your manuscript to PLOS ONE. After careful consideration, we feel that it has merit but does not fully meet PLOS ONE’s publication criteria as it currently stands. Therefore, we invite you to submit a revised version of the manuscript that addresses the points raised during the review process.

We would appreciate receiving your revised manuscript by Dec 05 2019 11:59PM. To enhance the reproducibility of your results, we recommend that if applicable you deposit your laboratory protocols in protocols.io, where a protocol can be assigned its own identifier (DOI) such that it can be cited independently in the future. For instructions see: http://journals.plos.org/plosone/s/submission-guidelines#loc-laboratory-protocols

We look forward to receiving your revised manuscript.

Kind regards,

Juan J Loor

Academic Editor

PLOS ONE

Reviewers' comments:

Reviewer's Responses to Questions

**Comments to the Author**

1. If the authors have adequately addressed your comments raised in a previous round of review and you feel that this manuscript is now acceptable for publication, you may indicate that here to bypass the “Comments to the Author” section, enter your conflict of interest statement in the “Confidential to Editor” section, and submit your "Accept" recommendation.

Reviewer #1: (No Response)

Reviewer #2: (No Response)

2. Is the manuscript technically sound, and do the data support the conclusions?

Reviewer #1: Partly

Reviewer #2: Yes

3. Has the statistical analysis been performed appropriately and rigorously? 

Reviewer #1: Yes

Reviewer #2: Yes

4. Have the authors made all data underlying the findings in their manuscript fully available?

Reviewer #1: Yes

Reviewer #2: Yes

5. Is the manuscript presented in an intelligible fashion and written in standard English?

Reviewer #1: Yes

Reviewer #2: Yes

6. Review Comments to the Author

Reviewer #1: the insistence that brachycephalic obstructive airway syndrome is best evaluated by direct visual examination misses the facts that a. this is a functional disease b. there is no current scheme to score CT/ endoscopy lesions with disease severity. Please adjust lines 59 and 319 -321 accordingly.

I think the authors are stating that there is a relationship between bfs and bs but they do not clearly define bs. From the study it seems that bs is a composite of various external conformational measurements. If this is the case can this be clearly stated.

Can the authors address the weak r results

Line 277. this study does not 'establish' - there is no validation of the bfs. preferable to use 'introduce' to avoid over confidence in an as yet unproven system. This is particularly the case as a new (and interesting) decibel recording system has been used which needs further investigation before it is advocated as a grading scheme

Reviewer #2: The title, although improved, remains in need of editing. The intent of the title is to convey that through concerted selection practices targeting BS can reduce the prevalence of BS. The current title of “Breeding French bulldogs with unobstructed breathing – possible but a long way to go” while catchy is still not informative. Of course it is possible to breed French bulldogs with unobstructed breathing. That however is not the intent of the manuscript. The manuscript is demonstrating that although French bulldogs with unobstructed breathing do exist, they are not that prevalent in the population the authors used and to increase the proportion of French bulldogs in a population that do not have BS it will take concerted selection and time in order to maintain genetic diversity. A more accurate title would be something along the lines of that conveys the concept that concerted selection can reduce obstructed breathing in the French bulldog but will take time.

Line 78 “Exercise testing is used both in humans [7, 8] and dogs [9] to measure reduced functional ability.” Please define what is meant by “functional ability”.

Line 164 should be microsatellites.

Line 268-269. The authors use reference 9 to illustrate the effect of BS on health and welfare. However the study detailed in reference 9 did not evaluate brachycephalic dogs. Thus this statement still is in need of references to demonstrate that “several studies….”

Line 268-269: Additionally the sentence needs editing as it appears something is missing: “..several studies have been performed to evaluate the severity of BS.” It would appear that the authors are tying together severity of BS and an impact on health or function.

Line 272: should be reflect the severity of BS (not reflect upon).

Line 272-273: comma needed after characterization “To lend support to the functional characterization, efforts have also been made to identify the…”

Please check all references. Line 286 reference for pure breed genetic variation is the R program for analysis

7. PLOS authors have the option to publish the peer review history of their article (what does this mean?). If published, this will include your full peer review and any attached files.

Reviewer #1: No

Reviewer #2: No

---

## [Author Response · Author response to Decision Letter 1]

15 Nov 2019

6. Review Comments to the Author

Reviewer #1: the insistence that brachycephalic obstructive airway syndrome is best evaluated by direct visual examination misses the facts that a. this is a functional disease b. there is no current scheme to score CT/ endoscopy lesions with disease severity. Please adjust lines 59 and 319 -321 accordingly.

I think the authors are stating that there is a relationship between bfs and bs but they do not clearly define bs. From the study it seems that bs is a composite of various external conformational measurements. If this is the case can this be clearly stated.

Thank you for this comment. We agree that some key definitions and explanations were missing:

In line 16-19 we have explained more precisely what BS is (ie. Brachycephalic syndrome (BS) is a pathophysiological disorder caused by excessive soft tissue within the upper airways of short-nosed dog breeds, causing obstruction of the nasal, pharyngeal and laryngeal lumen and thereby giving rise to various symptoms such as respiratory distress). In line 61 to 71 we have clarified why a functional assessment is a reasonable proxy measure for BS.

Can the authors address the weak r results

We have addressed the r results further in line 310-314

Line 277. this study does not 'establish' - there is no validation of the bfs. preferable to use 'introduce' to avoid over confidence in an as yet unproven system. This is particularly the case as a new (and interesting) decibel recording system has been used which needs further investigation before it is advocated as a grading scheme

We have rephrased this sentence (line 307-308): In the present study we have introduced the calculation of BSF scores, on a continuous scale, that rely on recordings of upper airway noise and decibel recordings before, during, and after exercise to assess the degree of BS

Reviewer #2: The title, although improved, remains in need of editing. The intent of the title is to convey that through concerted selection practices targeting BS can reduce the prevalence of BS. The current title of “Breeding French bulldogs with unobstructed breathing – possible but a long way to go” while catchy is still not informative. Of course it is possible to breed French bulldogs with unobstructed breathing. That however is not the intent of the manuscript. The manuscript is demonstrating that although French bulldogs with unobstructed breathing do exist, they are not that prevalent in the population the authors used and to increase the proportion of French bulldogs in a population that do not have BS it will take concerted selection and time in order to maintain genetic diversity. A more accurate title would be something along the lines of that conveys the concept that concerted selection can reduce obstructed breathing in the French bulldog but will take time.

We have changed the title so that it now reads: Breeding French bulldogs so that they breathe well - a long way to go.

Line 78 “Exercise testing is used both in humans [7, 8] and dogs [9] to measure reduced functional ability.” Please define what is meant by “functional ability”.

Exercise testing has been explained in more detail and with more appropriate references (see Line 104-112)

Line 164 should be microsatellites.

Spelling has been corrected (line 200)

Line 268-269. The authors use reference 9 to illustrate the effect of BS on health and welfare. However the study detailed in reference 9 did not evaluate brachycephalic dogs. Thus this statement still is in need of references to demonstrate that “several studies….”

We apologize for not providing sufficient documentation for this statement. Three more appropriate references have been provided (line 305 and 307)

Line 268-269: Additionally the sentence needs editing as it appears something is missing: “..several studies have been performed to evaluate the severity of BS.” It would appear that the authors are tying together severity of BS and an impact on health or function.

We have included ‘based on exercise tests’(line 307)to make the statement more comprehensible.

Line 272: should be reflect the severity of BS (not reflect upon).

We have removed ‘upon’ as suggested 

Line 272-273: comma needed after characterization “To lend support to the functional characterization, efforts have also been made to identify the…”

Thank you: we have included ‘,’ as suggested (line 316)

---

## [Editor Report · Decision Letter 2]

25 Nov 2019

Breeding French bulldogs with unobstructed breathing - possible but a long way to go

PONE-D-19-19251R2

Dear Dr. Fredholm,

We are pleased to inform you that your manuscript has been judged scientifically suitable for publication and will be formally accepted for publication once it complies with all outstanding technical requirements.

With kind regards,

Juan J Loor

Academic Editor

PLOS ONE
---

## [Editor Report · Acceptance letter]

6 Dec 2019

PONE-D-19-19251R2 

Breeding French bulldogs so that they breathe well - a long way to go 

Dear Dr. Fredholm:

I am pleased to inform you that your manuscript has been deemed suitable for publication in PLOS ONE. Congratulations! Your manuscript is now with our production department. 

With kind regards,

on behalf of

Dr. Juan J Loor 

Academic Editor

PLOS ONE